# *Spexin2* Is a Novel Food Regulator in *Gallus gallus*

**DOI:** 10.3390/ijms24054821

**Published:** 2023-03-02

**Authors:** Fengyan Meng, Yuping Wu, Yu Yu, Guixian Bu, Xiaogang Du, Qiuxia Liang, Xiaohan Cao, Anqi Huang, Xianyin Zeng, Linyan Huang, Fanli Kong, Yunkun Li, Xingfa Han

**Affiliations:** Isotope Research Lab, Biological Engineering and Application Biology Department, Sichuan Agricultural University, Ya’an 625014, China

**Keywords:** *Spexin2*, appetite regulation, hypothalamus, chicken

## Abstract

*Spexin2* (*SPX2*), a paralog of *SPX1*, is a newly identified gene in non-mammalian vertebrates. Limited studies in fish have evidenced its important role in food intake and energy balance modulation. However, little is known about its biological functions in birds. Using the chicken (c-) as a model, we cloned the full-length cDNA of *SPX2* by using RACE-PCR. It is 1189 base pair (bp) in length and predicted to generate a protein of 75 amino acids that contains a 14 amino acids mature peptide. Tissue distribution analysis showed that *cSPX2* transcripts were detected in a wide array of tissues, with abundant expression in the pituitary, testis, and adrenal gland. *cSPX2* was also observed to be ubiquitously expressed in chicken brain regions, with the highest expression in the hypothalamus. Its expression was significantly upregulated in the hypothalamus after 24 or 36 h of food deprivation, and the feeding behavior of chicks was obviously suppressed after peripheral injection with cSPX2. Mechanistically, further studies evidenced that cSPX2 acts as a satiety factor via upregulating cocaine and amphetamine regulated transcript (*CART*) and downregulating agouti-related neuropeptide (*AGRP*) in hypothalamus. Using a pGL4-SRE-luciferase reporter system, cSPX2 was demonstrated to effectively activate a chicken galanin II type receptor (cGALR2), a cGALR2-like receptor (cGALR2L), and a galanin III type receptor (cGALR3), with the highest binding affinity for cGALR2L. Collectively, we firstly identified that *cSPX2* serves as a novel appetite monitor in chicken. Our findings will help clarify the physiological functions of *SPX2* in birds as well as its functional evolution in vertebrates.

## 1. Introduction

Chicken is a popular domestic poultry species and its appetite regulation is closely associated with food consumption and growth performance [1]. However, feed intake is a complex process influenced by both individual and environmental variables [2]. These signals are mainly received and integrated in the hypothalamus, wherein the proopiomelanocortin (POMC), neuropeptide Y (NPY), and agouti-related peptide (AGRP) neurons respond to release their corresponding neuropeptides, and subsequently regulate feeding behavior and energy expenditure [3]. Over the last three decades, research in this field has largely improved our understanding of the appetite regulation with the ongoing identification of new appetite regulatory peptides, such as cocaine and amphetamine regulated transcript (CART) [4], orexin [5], and ghrelin [6]. Further investigation and the identification of novel food intake regulators will not only shed light on the appetite regulation mechanisms, but also provide potential strategies in dealing with abnormal food consumption in chickens [1,7].

Spexin (SPX), a novel neuropeptide composed of 14 amino acids, is highly conserved from fish to mammals [8], suggesting its essential physiological roles in vertebrates. Consistent with its wide tissue distribution, *SPX* has been confirmed to participate in multiple physiological functions, such as food intake [9,10,11,12,13,14], reproduction [15,16,17,18], and anxiety [19,20]. It is worth noting that *SPX* can serve as a satiety factor universally in mice [11,12], chickens [13,14,21], goldfish [9], zebrafish [10], and Siberian sturgeon [22], implicating that its function of food regulation is well-conserved across vertebrates. Further research shows that *SPX* suppresses appetite through the modulation of central orexigenic and anorexigenic signals, such as NPY, AgRP, and POMC [9,12,21,23]. SPX, as a tetradecapeptide, exerts its biological functions via activating its cell-surface receptors, such as galanin receptor 2 (GALR2) and GALR3, but not GALR1 [8,16]. Interestingly, apart from the above, two more GALR-like receptor (i.e., GALR1L and GALR2L) subtypes were identified to exist in chicken [24,25]. Our recent studies showed that chicken (c-) SPX (cSPX) had the highest affinity for GALR2L to stimulate the MAPK/ERK cascade in chickens [21]. These studies strongly evidenced that GALR2L is a key functional receptor of SPX as well.

Recently, an *SPX* paralog, namely *SPX2*, has been identified in non-mammalian vertebrates, therefore the original *SPX* is now designated as *SPX1* [8,26,27]. Compared with *SPX1*, little information is available regarding either the expression pattern or the physiological roles of *SPX2*. To the best of our knowledge, so far there are only a few studies regarding *SPX2* which are all limited to fish. Similar to SPX1, zebrafish SPX2 is also able to specifically activate human, *Xenopus*, and zebrafish GALR2 or GALR3 [8], suggesting that the two different peptides possibly share some similarity in their physiological functions. Using whole-mount in situ RNA hybridization, *SPX2* was evidenced to be restrictively expressed in the hypothalamic preoptic area of zebrafish [28]. Subsequently, using qRT-PCR, *SPX2* was detected to be widely expressed in tissues including the central nervous system and peripheral tissues [26,27], with high expression in testes and the brain [27]. Intriguingly, the expression of *SPX2* in hypothalamus was altered under different feeding states in zebrafish [27], implying its potential role in feeding control. Indeed, intraperitoneal injection of SPX2 inhibited food intake, while the disruption of this gene via TALENs markedly increased food consumption in zebrafish [27]. Similarly, the brain *SPX2* was also reported to be regulated by food deprivation in half-smooth tongue sole [26], reinforcing the concept that *SPX2* is a novel moderator of appetite. However, whether *SPX2* has similar function in birds is unknown.

Unlike *SPX1*, the functioning of *SPX2* has not yet been fully elucidated, and the prior research has only focused on fish. Though *SPX2* has been verified to exist in chickens, its gene structure, expression pattern, and functionality are all largely unknown. Thus, in this study, we aimed to clone the full-length cDNA of *SPX2* and investigate its tissue distribution and potential functions, especially appetite regulation in chickens.

## 2. Results

### 2.1. Cloning and Sequence Analysis of Chicken (c-) SPX2

According to the partial cDNA sequence of *cSPX2* (GenBank accession No.: KF601213) deposited in GenBank, we cloned its full-length cDNA from an adult chicken brain using RT- and RACE-PCR. As shown in Figure 1A, the cloned *cSPX2* has 1189 nucleotide bases, and its 5′-untranslated region (5′-UTR), CDS, and 3′-UTR are 293 bp, 228 bp, and 668 bp, respectively, in length. It was predicted to encode a precursor protein of 75 amino acids (aa), which could generate a 14-aa mature peptide after proteolytic processing at putative cleavage sites (R33, G48R49R50). Comparison of the *cSPX2* cDNA sequence with chicken genomes (https://asia.ensembl.org/Gallus_gallus/Info/Index (accessed on 8 January 2023), *cSPX2* consists of four exons separated by three introns (Figure 1B).

Sequence alignment revealed that the precursors of SPX2 shares high sequence similarity in birds, but comparatively lower identities among different other vertebrate classes, with an obvious sequence variation noted at its NH_2_-terminal and COOH-terminal region (Figure 2). It is noteworthy that cSPX2 mature peptide displays a high degree of amino acid sequence identity to that of ring-necked pheasant (100%), rock ptarmigan (100%), greater sage-grouse (100%), green anole (92.86%), *Xenopus laevis* (78.57%), coelacanth (92.86%), and zebrafish (78.57%). Despite the fact that the sequence of cSPX2 mature peptide is four amino acids different from cSPX1, they are highly conserved (64.29%).

### 2.2. Synteny Analysis

To confirm whether the cloned *cSPX2* is orthologous to the gene identified in other vertebrate species, we performed synteny analysis by searching its neighboring genes in the genomic regions of chicken and other vertebrate species. As shown in Figure 3, *SPX2* is located on a syntenic region conserved in chicken, green anole, *Xenopus laevis*, coelacanth, and zebrafish, suggesting the cloned gene is an ortholog of *SPX2*. In addition, *SPX2* and *GAL* are on the same chromosome in these species. However, *SPX2* is absent in mammals, such as humans and mice.

### 2.3. Tissue Distribution of SPX2 in Adult Chickens

To explore the physiological functions of *SPX2* in chickens, we detected its tissue distribution in adult chickens using qRT-PCR. As shown in Figure 4, *cSPX2* was widely expressed in all tissues except gizzard. It was abundantly expressed in the anterior pituitary, testis, and adrenal gland, and moderately expressed in the jejunum, ileum, and pancreas. In the central nervous system (CNS), *cSPX2* was found to be ubiquitously expressed in the spinal cord and all the brain regions examined, with the highest expression in the hypothalamus.

### 2.4. Effects of cSPX2 on Chicken Feed Intake

The mature peptide of cSPX2 shares high sequence similarity with cSPX1, which can act as a satiety factor in chicken [14,21]. We thus speculated that cSPX2 also has the anorexigenic properties. In order to verify this hypothesis, we detected its transcriptional response to fasting in the hypothalami of two-week-old chicks. As shown in Figure 5A, 24- or 36-h fasting could obviously elevate (*p* < 0.01) the levels of *cSPX2* mRNA in the chick hypothalamus. Then, the feed consumption was measured after cSPX2 injection. The cumulative feed intake of chicks did not alter (*p* > 0.05) by three different doses of cSPX2 after 6 h treatment, but it was effectively inhibited (*p* < 0.05) at 12 h post-injection (Figure 5B). Moreover, after 24 h post-injection, a moderate (50 ng/g BW) and a high (100 ng/g BW) dosage of cSPX2 could still markedly decrease (*p* < 0.05) the food intake in chicks.

### 2.5. Transcriptional Regultion of Appetite-Regulating Regulators by cSPX2

To elucidate the mechanism of cSPX2 in feeding regulation, we monitored the expression of appetite-regulating factors in the hypothalamus. As shown in Figure 6A, cSPX2 could visibly inhibit (*p* < 0.05) *AGRP* mRNA levels at 6 h post-injection. Oppositely, the expression of a pro-melanin-concentrating hormone (*PMCH*) was stimulated (*p* < 0.05) by cSPX2 administration. *CART* mRNA levels exhibited downward tendencies, and no significant difference was examined after 6 h injection with cSPX2. Conversely, transcripts of *CART* in hypothalami of three-week-old chicks were obviously elevated (*p* < 0.05) after cSPX2 administration for 24 h, and the expressions of *NPY*, *AGRP*, *POMC*, and *PMCH* were not regulated (*p* > 0.05) by cSPX2 (Figure 6B).

### 2.6. Potency of cSPX2 in Activating Chicken Galanin Receptors

To investigate whether cSPX2 is a natural ligand for galanin receptors in chickens, each receptor was transfected into HEK293 cells, and subjected to peptide treatment. Receptor activation was examined by pGL4-SRE-luciferase reporter system. As shown in Figure 7, cSPX2 could not efficiently stimulate luciferase activity in HEK293 cell expressing cGALR1 (accession No.: NP_001121534) and cGALR1L (accession No.: NP_001121533). On the contrary, cGALR2 (accession No.: ACB22016), cGALR2L (accession No.: ACD99708) and cGALR3 (accession No.: ACE60644) could be initiated by cSPX2 in a dose-dependent manner, and the EC_50_ values are 218 nM, 53 nM, and 110 nM, respectively. Among them, cSPX2 exhibited the highest binding ability with cGALR2L.

## 3. Discussion

As a paralog of *SPX1*, *SPX2* has been discovered in non-mammalian species [8,26,27], but little is known about its physiological functions. In this study, using chicken as an animal model, we firstly cloned the full-length cDNA encoding *SPX2*. Synteny analysis revealed that it is orthologous to the *SPX2* of anole lizards, *Xenopus*, coelacanths, and zebrafish. Concurring with the previous report that *SPX2* is found in a variety of vertebrate species but not in mammals [8,26,27]. We found that *cSPX2* and *cGAL* are closely localized on the same chromosome and possibly coevolutionary in vertebrates, as reported previously [8]. Similar to *SPX1*, *cSPX2* can generate a 14-aa mature peptide flanked by cleavage sites, and the mature peptide is highly conserved, suggesting SPX2 may play similar important physiological roles as SPX1 in non-mammalian vertebrates.

Tissue distribution analysis revealed that *cSPX2* mRNA was widely detected in various tissues of a chicken, which is similar with the tissue expression profiles reported in both tongue sole [26] and zebrafish [27]. Remarkably, we found that in chickens the pituitary had the highest expression of *cSPX2*. As a master hormone secretion center, the pituitary gland controls metabolism, growth, sexual maturation, and reproduction as well as many other vital physiological functions and processes [29]. Therefore, it is likely that the pituitary will operate as a key mediator for SPX2 to regulate various physiological processes in non-mammalian vertebrates. Indeed, it was reported that an intraperitoneal injection of SPX2 could largely alter the expression of the gonadotropin α subunit, the luteinizing hormone β (LHβ), and the follicle-stimulating hormone β subunit (FSHβ) in the pituitary in half-smooth tongue sole [30], and that the knockout of *SPX2* could greatly promote body growth in zebrafish by enhancing the expression of pituitary growth hormone (GH) [27]. High amounts of *cSPX2* mRNA were also observed in chicken testes and adrenal glands, suggesting it may also directly regulate the functions of these organs, such as the sexual/adrenal steroid secretion, spermatogenesis, etc. Of note, *cSPX2* was also moderate in various brain regions, with the highest expression in the hypothalamus, suggesting that *SPX2* may be involved in the regulation of food intake in chickens. Taken together, *SPX2* is widely distributed in chickens and potentially exerts pleiotropic functions. To better understand its pleiotropic functions in each individually expressed tissue of chickens, more extensive and in-depth investigations are required.

To explore the putative roles of *cSPX2* in feeding control in chickens, we firstly detected its hypothalamic mRNA expression in response to food deprivation. Consequently, fasting remarkably increased *cSPX2* expression in the hypothalamus, which is analogous to findings reported in half-smooth tongue sole [26]. Conversely, in zebrafish *SPX2* mRNA was downregulated by fasting [27]. Despite discrepancies, changes in *SPX2* expression under different feeding status conditions across species suggest its potential role in appetite regulation. To further validate its putative roles in feeding control, we performed in vivo SPX administration in chicks. Consequently, SPX2 administration greatly decreased the food intake of chicks, and similar findings were also reported in zebrafish [27]. Fish with *SPX2* knockout also exhibited higher food consumption than WT fish [27]. All these results strongly suggest that, like SPX1, SPX2 acts as a novel appetite regulator in chickens as well as some other non-mammalian vertebrates.

To elucidate the underpinning mechanism of cSPX2 in appetite regulation, we further detected its regulatory influence on the expression of key feeding control factors in the hypothalamus of chicks. Following intravenous cSPX2 injection, *AGRP*, a well-known orexigenic gene, was visibly downregulated at 6 h post injection, whereas the anorexigenic factor, *CART*, was notably upregulated at 24 h post injection. In parallel, the cumulative food consumption of chicks were substantially reduced onward from 12 h post injection. Consistently, both in vivo SPX2 injection and *SPX2* knockout evidenced that SPX2 controls food intake in zebrafish by regulating hypothalamic *AGRP* and *CART* expression as well [27]. Unexpectedly, the expression of *PMCH*, a potential orexigenic factor [31], was increased after cSPX2 treatment for 6 h, similar to what was seen with SPX1 in goldfish [8]. Thus, it appeared that SPXs (both SPX1 and SPX2) could exert an early and quick action on the expression of *PMCH* in the hypothalamus. However, this early and quick action of SPX2 on *PMCH* expression cannot essentially change the pattern of chicks’ decreased food consumption. SPX2-induced changes of *AGRP* and *CART* were still dominant over PMCH to control food intake. Therefore, in chickens, SPX2 acts as a satiety factor to inhibit food intake mainly by downregulating *AGRP* and upregulating *CART* in the hypothalamus.

SPX2 exerts its physiological functions by activating its receptors. Previous studies indicated that SPX2 could elevate SRE-luc activity in cells expressing GALR2 and GALR3, but not in cells expressing GALR1 [8,16]. A little surprisingly, in the present study we found cSPX2 could not only activate cGALR2 and cGALR but also cGALR2L. Among them, cSPX2 exhibited the highest binding ability with cGALR2L. Accordingly, it is likely that cSPX2 exerts its feed regulation mainly through binding and activating cGALR2L. Interestingly, cSPX1 [21], but not cSPX2 could inhibit the food intake of chicks after 6 h post-injection at the same treatment dosage, suggesting that cSPX2 is potentially not as effective as cSPX1 in inhibiting the appetite of chicks. This is possibly attributed to the differential potency of SPX1 and SPX2 in activating galanin receptors. In both chickens [21] and humans [8] it has been evidenced that SPX1 has a higher potential to activate GALR2 and GALR3 than SPX2.

In summary, we cloned the full-length cDNA of chicken *SPX2* and investigated its expression patterns and functionality. *cSPX2* can generate a conserved mature peptide with 14-aa, which can activate GALR2, GALRL2L, and GALR3. Tissue distribution analyses show that *cSPX2* is widely expressed in chickens, with abundant expression in the pituitary, testis, and adrenal glands, and its expression in hypothalamus was upregulated by fasting. In vivo administration of cSPX2 could effectively inhibit the feed intake of chickens, by modifying the expression of hypothalamic *AGRP* and *CART*. Collectively, our results firstly evidenced that cSPX2 serves as a novel appetite monitor in chickens, which will facilitate the understanding of the physiological functions of SPX2 in birds as well as its functional evolution in vertebrates.

## 4. Materials and Methods

### 4.1. Chemicals, Peptides and Primers

All chemicals were purchased from Sigma-Aldrich (Shanghai, China) unless stated otherwise. Chicken SPX2 mature peptide (cSPX2, NWGPQSILYLKGRY-NH_2_) with amidated C-terminus was synthesized by GL Biochem Ltd. (Shanghai, China), and its structure was verified by mass spectrometry. Dulbecco’s Modified Eagle Medium (DMEM), fetal bovine serum (FBS), and penicillin-streptomycin were all obtained from Hyclone (Cytiva, Shanghai, China). JetPRIME^®^ transfection reagent was purchased from Polyplus (Illkirch, France). All primers used in this study were synthesized by Chengdu Tsingke Biotechnology and listed in Appendix A.

### 4.2. Animals and Samples

Adult chickens (3 males and 3 females, 31-week-old) or chicks (Lohmann layer) used in the present study were obtained from Muxing company in Sichuan, China. Chickens were sacrificed to collect various tissues including different brain regions. All samples were snap-frozen in liquid nitrogen and then stored at −80 °C before RNA extraction. All procedures involving animals were approved by the Ethics Committee of Sichuan Agricultural University.

### 4.3. RNA Extraction and Reverse Transcription (RT)

Total RNA was isolated from various chicken tissues with RNAzol (Molecular Research Center, Cincinnati, OH, USA) according to the manufacturer’s instructions, and then 1 μg of total RNA was used as template to reverse transcription by using the PrimeScript^®^ RT reagent kit with gDNA eraser (Takara, Dalian, China). The generated cDNA samples were diluted and then used as template for subsequent PCR or quantitative real-time PCR (qRT-PCR) for detecting the expressions of target genes.

### 4.4. Cloning the Full-Length cDNA of cSPX2

According to the predicted partial coding sequence (CDS) of *cSPX2* (GenBank No.: KF601213), the gene-specific primers were designed to amplify the 5′-cDNA and 3′-cDNA ends by using SMART-RACE cDNA amplification kit (Clontech, Palo Alto, CA, USA). The amplification products were detected by agarose gel electrophoresis, and then ligated into pTA2 vector (Toyobo, Shanghai, China) for sequencing.

### 4.5. Functional Characterization of cSPX2 and Galanin Receptors

The potency of cSPX2 on activating chicken galanin receptors was evaluated by pGL4-SRE-Luciferase system according to our previously established methods [21]. Using JetPRIME^®^ transfection reagent, HEK293 cells were, respectively, transfected by five galanin receptor subtypes, including GALR1, GALR1L, GALR2, GALR2L, and GALR3, and then were treated with different doses of cSPX2 for an additional 6 h. Whereafter, 1× Passive lysis buffer (Promega, Beingjing, China) was added into cells and the luciferase activity was determined with the luciferase assay kit (Promega).

### 4.6. Quantitative Real-Time PCR (qRT-PCR)

According to our previously established method [21], qRT-PCR was conducted on the CFX96 Real-time PCR Detection System (Bio-Rad, Hercules, CA, USA), and PCR procedures consisted of 40 cycles of 94 °C for 20 s, annealing for 15 s, 30 s extension at 72 °C. To assess the specificity of PCR amplification, melting curve analysis was performed at the end of the reaction. Finally, the mRNA levels of target genes were normalized against β-actin, calculated with the 2^−ΔΔCT^ method, and then expressed as the fold change relative to the chosen tissue or control. Primer sequences of target and reference genes are shown in Appendix A.

### 4.7. Effects of cSPX2 on Chick Feed Intake

Given that *cSPX1* plays a significant role in appetite regulation, its paralog, *cSPX2*, may perform a similar physiological function. Thus, starvation experiment was conducted in this study. Two-week-old male chicks were divided equally into three groups (*n* = 24): one control group and two experimental groups. Chicks in control group were fed at regular times, and chicks in the two experimental groups were fasted for 24 and 48 h, respectively. After fasting, chicks were sacrificed and hypothalami were collected for determining *cSPX2* expression response to fasting.

To further investigate the effect of *cSPX2* on feed intake, similar-weight chicks (~180 g/chick) were randomly assigned to four groups (*n* ≥ 8), and then were deprived of food for 12 h. Then, chicks in control group were intravenously injected with PBS, and chicks in the three experimental groups were injected with 10 ng/g, 50 ng/g and 100 ng/g body weight of cSPX2, respectively. After that, chicks were re-fed with the pre-weighed diet, and the residual diets were collected and weighed at 6 h, 12 h, and 24 h post-injection, respectively. Food consumption was counted by the total input of food minus the leftover.

### 4.8. Transcriptional Regulation of Appetite-Regulating Factors by cSPX2

Three-week-old male chicks were randomly divided into two groups (*n* ≥ 4), and then were intravenously injected with PBS (control) or cSPX2 (50 ng/g body weight). After 6 h of administration, the chick hypothalami were collected, and the mRNA expression levels of neuropeptide Y (*NPY*), agouti gene-related protein (*AGRP*), cocaine- and amphetamine-regulated transcript (*CART*), proopiomelanocortin (*POMC*), and pro-melanin-concentrating hormone (*PMCH*) were detected by qRT-PCR. We also evaluated the expression of the above appetite-related genes in the chick hypothalamus after 24 h injection with cSPX2 by using the same treatment procedure.

### 4.9. Data Analysis

All statistical analysis was performed with GraphPad Prism 9 software (La Jolla, CA, USA). The two-tailed Student’s *t* test was conducted to compare two groups. For comparison between more than two groups, a one-way ANOVA followed by Turkey’s test was used. Data were expressed as means ± SEM, and statistical significance was set at *p* < 0.05.

## Figures and Tables

**Figure 1 ijms-24-04821-f001:**
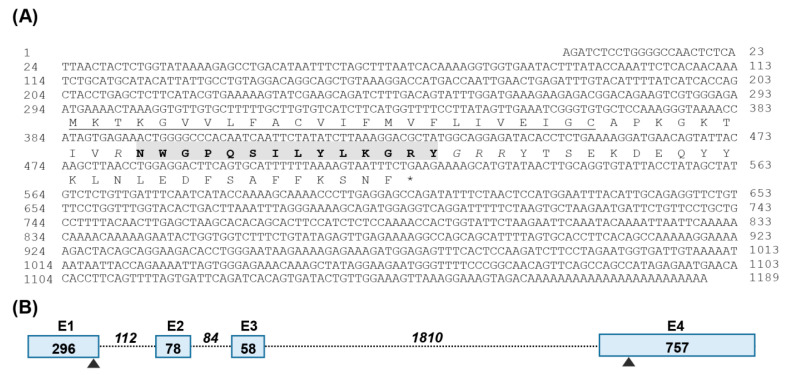
Cloning and sequence analysis of chicken (c-) *SPX2*. (**A**) The full-length cDNA sequence and deduced amino acid of chicken *Spexin2* (*cSPX2*). The predicted signal peptide (24 amino acids) is underlined, and the mature peptide (14 amino acids) is shaded in gray, which is flanked by two dibasic cleavage sites noted by italic letters. (**B**) Exon (E)-intron organization of *cSPX2*, the numbers indicate the sizes of exons or introns, and arrowheads represent the locations of start codon and stop codon.

**Figure 2 ijms-24-04821-f002:**
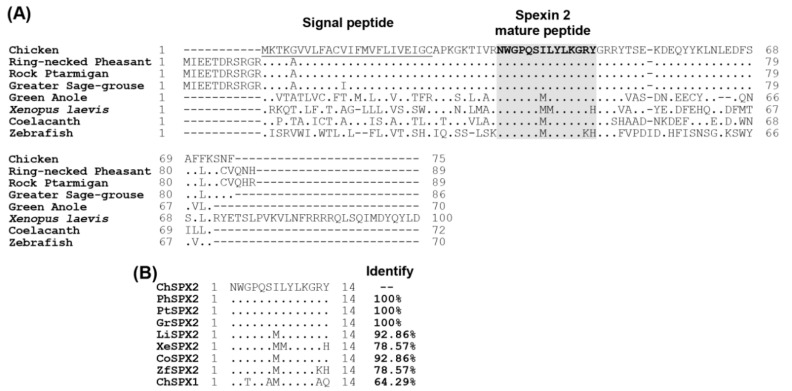
Amino acid sequence alignment of SPX2. (**A**) Multiple amino acid sequence alignments of chicken SPX2 with those of other species, including ring-necked pheasant (XM_031608967), rock ptarmigan (XM_048949770), greater sage-grouse (XM_042813209), green anole (KF601212), *Xenopus laevis* (XM_041589573), coelacanth (KF601216), and zebrafish (KF601217). The signal peptide is underlined and the mature peptides are shaded in gray. (**B**) Sequence conservation between chicken Spexin 2 mature peptide (ChSPX2) with ring-necked pheasant (PhSPX2), rock ptarmigan (PtSPX2), greater sage-grouse (GrSPX2), green anole (LiSPX2), *Xenopus laevis* (XeSPX2), coelacanth (CoSPX2), zebrafish (ZfSPX2), and chicken Spexin 1 mature peptide (ChSPX1). The identical amino acids are denoted by dots.

**Figure 3 ijms-24-04821-f003:**
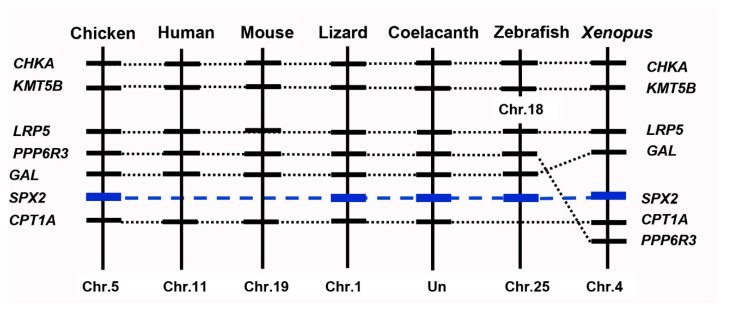
Conserved synteny for the genomic location of chicken *SPX2* with its orthologous gene in other vertebrates. Based on existing genome assemblies, *SPX2* is absent in humans and mice. Dotted lines indicate the syntenic genes, and dashed lines denote the location of *SPX2*. Note: genome assemblies of Chicken (GRCg7w), Human (GRCh38.p13), Mouse (GRCm39), Lizard (AnoCar2.0v2), Coelacanth (LatCha1), Zebrafish (GRCz11), and Xenopus (UCB_Xtro_10.0) as well as their gene notation information from Ensembl were used in the analysis.

**Figure 4 ijms-24-04821-f004:**
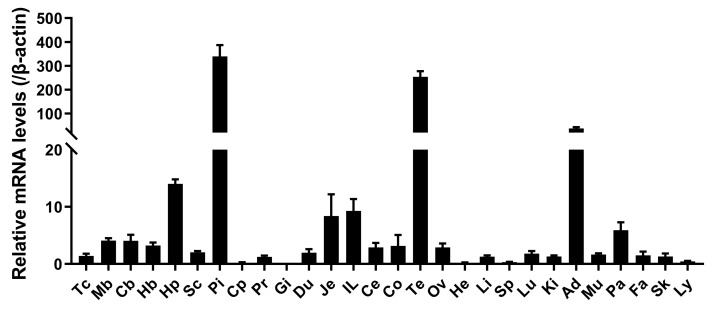
The expression profiles of *SPX2* in adult chickens. Quantitative real-time PCR assay of *cSPX2* in chicken various tissues, including the telencephalon (Tc), midbrain (Mb), cerebellum (Cb), hindbrain (Hb), hypothalamus (Hp), spinal cord (Sc), anterior pituitary (Pi), crop (Cp), proventriculus (Pr), gizzard (Gi), duodenum (Du), jejunum (Je), ileum (IL), caecum (Ce), colon (Co), testes (Te), ovary (Ov), heart (He), liver (Li), spleen (Sp), lung (Lu), kidney (Ki), adrenal gland (Ad), muscle (Mu), pancreas (Pa), abdominal fat (Fa), skin (Sk), and lymph (Ly). The mRNA levels of *cSPX2* were normalized to that of *β-actin* and expressed as the fold difference compared with that of telencephalon (Tc). Each data point represents the mean ± SEM of 6 adult chickens (*n* = 6).

**Figure 5 ijms-24-04821-f005:**
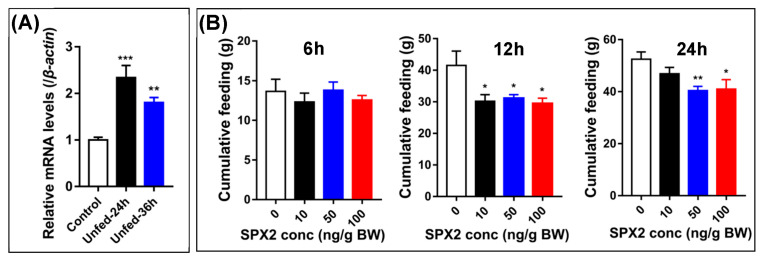
Effects of cSPX2 on chicken food intake. (**A**) Fasting could induce *cSPX2* mRNA expression in chick hypothalamus (*n* = 8). (**B**) Food consumption of chicks after intravenous injection of cSPX2 for 6 h, 12 h, and 24 h. Each data point represents the mean ± SEM (*n* ≥ 8). * *p* < 0.05, ** *p* < 0.01, *** *p* < 0.001 vs. respective control.

**Figure 6 ijms-24-04821-f006:**
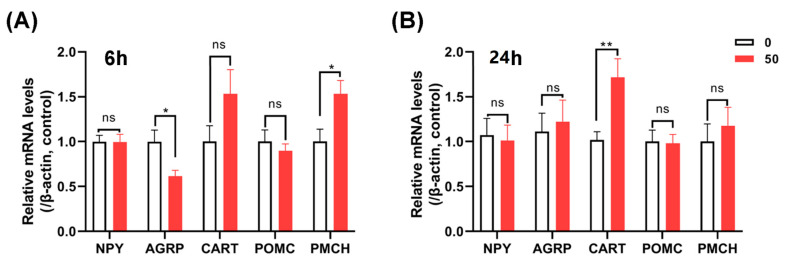
Perineural injection of cSPX2 on the expressions of orexigenic and anorexigenic genes in chick hypothalamus. Three-week-old chicks were intravenously injected with cSPX2 (50 ng/g BW) or PBS (as control). The hypothalami were harvested at (**A**) 6 and (**B**) 24 h post-injection, and used to detect the transcript expressions of *AGRP*, *NPY*, *CART*, *POMC*, and *PMCH*. Each data point represents the mean ± SEM (*n* ≥ 4). * *p* < 0.05, ** *p* < 0.01, ^ns^
*p* > 0.05 vs. control.

**Figure 7 ijms-24-04821-f007:**
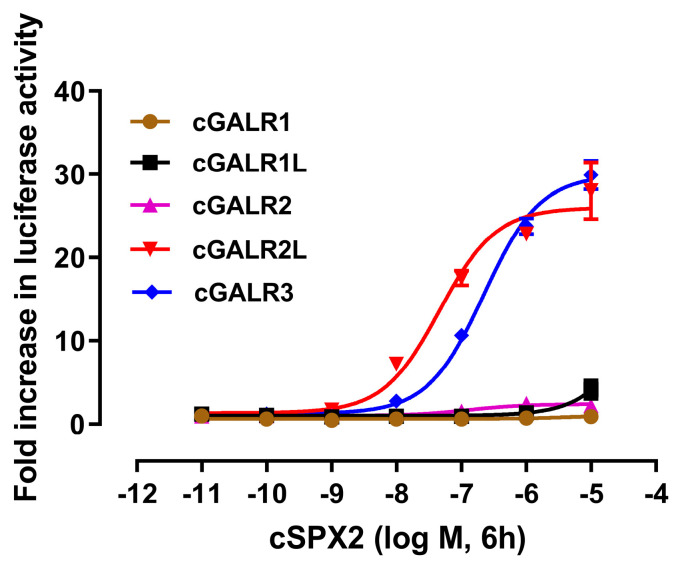
The effects of cSPX2 (10^−11^–10^−5^ M, 6 h) on activating chicken galanin receptors expressed in HEK293T cells, monitored by pGL4-SRE-luciferase reporter system. Each data point represents mean ± SEM of three replicates (*n* = 3).

## Data Availability

All data generated or analyzed during this study are included in this published article.

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
