# Peer review of "Spexin2 Is a Novel Food Regulator in Gallus gallus"

_ijms, 2023, doi:10.3390/ijms24054821_

Round 1

Reviewer 1 Report

Summary:

This manuscript describes the initial functional characterization of a novel gene found in non-mammalian vertebrates. The expression of this gene across multiple tissues is tested, its receptors characterized, and its role in feeding behavior investigated. This is an important work in understanding how appetite is regulated in animals but revisions of the manuscript are required to ensure that these results a clearly described in a way that maximizes the impact of this research.

Review:

The chicken Spx partial CD described in this manuscript (KF601213) is linked to NCBI Gene ID: 101751123 (LOC101751123 uncharacterized LOC101751123). The focus of this manuscript is the functional characterization of this poorly described gene, but the reader cannot even be certain precisely which gene is being described. Please add the specific gene accession to the manuscript. Moreover, this manuscript represents the first functional characterization for this gene and will be used to assign standardized gene nomenclature across birds. With this in mind, "c" or "ch" should not be used to reference chicken genes (see PMID: 19607656 and http://birdgenenames.org/cgnc/guidelines.jsp). Moreover, naming this gene Spx2 is problematic. Humans have SPX1 spexin hormone (HGNC:28139) but SLC37A2 (HGNC:20644) already has the alias "SPX2", SLC37A3 (HGNC:20651) has an alias SPX3 and SLC37A4 (HGNC:4061) has the alias SPX4. Each of these human genes have well defined vertebrate orthologs, including in birds. Consider using Spx5 to ensure that the gene is clearly and unambiguously defined.

Standardized nomenclature should be used to refer to vertebrate genes, and where appropriate genes, mRNAs or proteins should be referenced with stable accessions from public databases (e.g., NCBI or Ensembl). For example, the name “cocaine and amphetamine-regulated transcript I (CART1)” is used in the manuscript but the human CART1 symbol was previously used for “cartilage paired-class homeoprotein 1”, which is now referred to as ALX1 ALX homeobox 1 (Gene ID: 8092), which has the ortholog in chicken Gene ID: 427871 ALX1. It is not clear if this is that same as the CART1 discussed in this manuscript. Likewise, melanin-concentrating hormone (MCH) could perhaps refer to PMCH pro-melanin concentrating hormone (Gene ID: 418091). Orexin and ghrelin are similar examples. Please update names to match standardized nomenclature and add accessions for all the sequences used in this study.

The new sequence of chicken SPX2 should be submitted to a sequence archive and the accession cited in this manuscript.

What controls are done to make sure that it is Spx2 amplified and not Spx1? Can Spx1 be amplified from the same tissues? Please cite the suppl table with primer sequences in the methods.

“Comparison of cSPX2 cDNA sequence with chicken genome (https://asia.en-89 sembl.org/Gallus_gallus/Info/Index)” – it is not clear what comparison was done and the link provided is a generic link to Ensembl rather than to a specific gene page.

The complement of genes for chicken will change based upon genome and genome version is being used. Specific information about the genome assembly (e.g., GRCg6a from red jungle fowl, GRCg7b from broiler, GRCg7w_WZ from white leghorn layer) and annotation (e.g., NCBI or Ensembl) information should be included for the synteny analysis. The gene order shown in the synteny analysis does not match what is displayed in Ensembl 109 bGalGal1.mat.broiler.GRCg7b and not all the genes identified on the figure can be identified (including Spx2). Without more detailed information this figure is not reproducible.

The age of the chicks/bird tissues used to test Spx2 expression across multiple tissues is not given. It would be useful information to know the age and sex where Spx2 is expressed.

Functional characterization of cSPX2 and galanin receptors: It is not clear precisely which galanin receptors were used in the transfection study (for example, no chicken GALR2L gene is currently found in NCBI or Ensembl). Please add accessions for the protein sequences used in this experiment. Moreover, why was a human cell line used (HEK293)? What evidence is there that GALR1, GALR1L & GALR2 were functional within the HEK293 cell line?

Author Response

Responds to the comments of Reviewer #1:

Q1: The chicken Spx partial CD described in this manuscript (KF601213) is linked to NCBI Gene ID: 101751123 (LOC101751123 uncharacterized LOC101751123). The focus of this manuscript is the functional characterization of this poorly described gene, but the reader cannot even be certain precisely which gene is being described. Please add the specific gene accession to the manuscript. Moreover, this manuscript represents the first functional characterization for this gene and will be used to assign standardized gene nomenclature across birds. With this in mind, "c" or "ch" should not be used to reference chicken genes (see PMID: 19607656 and http://birdgenenames.org/cgnc/guidelines.jsp). Moreover, naming this gene Spx2 is problematic. Humans have SPX1 spexin hormone (HGNC:28139) but SLC37A2 (HGNC:20644) already has the alias "SPX2", SLC37A3 (HGNC:20651) has an alias SPX3 and SLC37A4 (HGNC:4061) has the alias SPX4. Each of these human genes have well defined vertebrate orthologs, including in birds. Consider using Spx5 to ensure that the gene is clearly and unambiguously defined.

Response: KF601213 is the GenBank number (NCBI nucleotide database) rather than NCBI gene ID (NCBI gene database ID). Until today, chicken SPX2 is not found in NCBI gene database, so now there is still no NCBI gene ID for chicken SPX2. We have specified that “KF601213” is the the GenBank number of chicken SPX2 in the manuscript (see line 83; marked in red).

Dear reviewer, maybe, there are some misunderstandings. In the present study, we only used the letter "c" to stand for genes or proteins that belong to chickens rather than naming a gene or protein. For example, “injection with cSPX2” in the manuscript means injecting chicken SPX2 peptide. We (e.g., PMCID:PMC9673105; PMID:36774983) and others (e.g., PMC9170129) commonly used this writing style in the published papers.

Chicken Spexin2 (SPX2) has been named in the NCBI (GenBank No.: KF601213), So we did not name this gene in chickens, and we just used it according to the NCBI. The same gene was also named “SPX2” in half-smooth tongue sole (PMCID: PMC9376245) and zebrafish (PMID: 35293264, PMID: 30903017).

Q2: Standardized nomenclature should be used to refer to vertebrate genes, and where appropriate genes, mRNAs or proteins should be referenced with stable accessions from public databases (e.g., NCBI or Ensembl). For example, the name “cocaine and amphetamine-regulated transcript I (CART1)” is used in the manuscript but the human CART1 symbol was previously used for “cartilage paired-class homeoprotein 1”, which is now referred to as ALX1 ALX homeobox 1 (Gene ID: 8092), which has the ortholog in chicken Gene ID: 427871 ALX1. It is not clear if this is that same as the CART1 discussed in this manuscript. Likewise, melanin-concentrating hormone (MCH) could perhaps refer to PMCH pro-melanin concentrating hormone (Gene ID: 418091). Orexin and ghrelin are similar examples. Please update names to match standardized nomenclature and add accessions for all the sequences used in this study.

Response: We highly appreciated the reviewer’s good comments. Indeed, we searched the NCBI and found the names of partial genes were updated. According to your good suggestions, we have updated the names of genes with the newest official name from NCBI website and meanwhile added their corresponding accession number (see supplementary table 1) in the manuscript.

Q3: The new sequence of chicken SPX2 should be submitted to a sequence archive and the accession cited in this manuscript.

Response: According to suggestion, we have successfully submitted our newly cloned SPX2 sequence into the NCBI website, now it is still in line for acceptance. We will cite it in the manuscript once obtaining the accession number before publication.

Q4: What controls are done to make sure that it is Spx2 amplified and not Spx1? Can Spx1 be amplified from the same tissues? Please cite the suppl table with primer sequences in the methods.

Response: We cloned cSPX1 previously (PMCID: PMC9673105) and cSPX2 in this study. So, both their sequences were known. Conservation analysis shown that cDNA of the two genes just share 32.52% similarity. So using the specific primers to amplify the region which is with big difference between them can easily distinguish SPX2 from SPX1. Finally, by sequencing the qPCR products, we are 100% sure the qPCR amplification is SPX2 rather than SPX2. According to suggestion, we have cited suppl table with primer sequences in the methods (see line 323; marked in red).

Q5:“Comparison of cSPX2 cDNA sequence with chicken genome (https://asia.en-89 sembl.org/Gallus_gallus/Info/Index)” – it is not clear what comparison was done and the link provided is a generic link to Ensembl rather than to a specific gene page.

Response: Because, we firstly cloned cSPX2 in this study. So, now there is no a specific gene page for chicken SPX2 in Ensembl. We just blasted our cloned cSPX2 sequence against the chicken genome. That’s why we just provided the chicken genome link herein.  

Q6: The complement of genes for chicken will change based upon genome and genome version is being used. Specific information about the genome assembly (e.g., GRCg6a from red jungle fowl, GRCg7b from broiler, GRCg7w_WZ from white leghorn layer) and annotation (e.g., NCBI or Ensembl) information should be included for the synteny analysis. The gene order shown in the synteny analysis does not match what is displayed in Ensembl 109 bGalGal1.mat.broiler.GRCg7b and not all the genes identified on the figure can be identified (including Spx2). Without more detailed information this figure is not reproducible.

Response: According to suggestion, we have provided the genome assembly and annotation informations for all the genomes used in the studies (see line 128-131; marked in red).

We checked the Ensembl 109 bGalGal1.mat.broiler.GRCg7b, it seems to be just updated these days, which thus caused that “the synteny analysis result does not match what is displayed Ensembl 109 bGalGal1.mat.broiler.GRCg7b”. To avoid mistakes, we have deleted IGHMBP2 from the figure.   

 Q7: The age of the chicks/bird tissues used to test Spx2 expression across multiple tissues is not given. It would be useful information to know the age and sex where Spx2 is expressed.

Response: We have added the age of chicks/bird used according to suggestion (see line 289 marked in red for adult chickens; line 328 marked in red for chicks).

 Q8: Functional characterization of cSPX2 and galanin receptors: It is not clear precisely which galanin receptors were used in the transfection study (for example, no chicken GALR2L gene is currently found in NCBI or Ensembl). Please add accessions for the protein sequences used in this experiment. Moreover, why was a human cell line used (HEK293)? What evidence is there that GALR1, GALR1L & GALR2 were functional within the HEK293 cell line?

Response: According to suggestion, we have added the protein accession numbers for these galannin receptors in the revised manuscript (see line 187-189, marked in red).

Although HEK293 is a human cell line, it is one of the most widely used cell lines in research to characterize molecular functions & receptor signaling across species due to its high transfectivity, rapid growth rate, and ability to grow in a serum-free, suspension culture. So, we chose to use HEK293, just because it is a good and widely used cell line.

Special thanks to your good comments and suggestions!

Reviewer 2 Report

Comments to Authors 

           This study showed that cSPX2 serves as a novel appetite monitor in chicken.

           Spexin (SPX) is a highly conservative peptide hormone containing 14 amino acids and was discovered in 2007 by bioinformatics methods [1]. However, nothing is yet known about its role in the metabolism of birds, including broilers [1].

          Recently, it showed that SPX can act as a satiety factor by orchestrating the expression of key feeding regulators in the chicken hypothalamus but also help to facilitate a better understanding of its functional evolution across vertebrates [2].

          Authors are kindly requested to emphasize the current concepts about these issues in the context of recent knowledge and the available literature. This articles should be quoted in the References list.

References

1.      Effect of Fasting on the Spexin System in Broiler Chickens. Animals (Basel). 2021; 11 (2): 518. Published 2021 Feb 17. doi:10.3390/ani11020518.

2.      Characterization of spexin (SPX) in chickens: molecular cloning, functional analysis, tissue expression and its involvement in appetite regulation. Poult Sci. 2023; 102 (1): 102279. doi:10.1016/j.psj.2022.102279.

Author Response

Responds to the comments of Reviewer #2:

General Comment: This study showed that cSPX2 serves as a novel appetite monitor in chicken. Spexin (SPX) is a highly conservative peptide hormone containing 14 amino acids and was discovered in 2007 by bioinformatics methods [1]. However, nothing is yet known about its role in the metabolism of birds, including broilers [1]. Recently, it showed that SPX can act as a satiety factor by orchestrating the expression of key feeding regulators in the chicken hypothalamus but also help to facilitate a better understanding of its functional evolution across vertebrates [2]. Authors are kindly requested to emphasize the current concepts about these issues in the context of recent knowledge and the available literature. This articles should be quoted in the References list.  References:

  1. Effect of Fasting on the Spexin System in Broiler Chickens. Animals (Basel). 2021; 11 (2): 518. Published 2021 Feb 17. doi:10.3390/ani11020518.
  2. Characterization of spexin (SPX) in chickens: molecular cloning, functional analysis, tissue expression and its involvement in appetite regulation. Poult Sci. 2023; 102 (1): 102279. doi:10.1016/j.psj.2022.102279.

Response: We highly appreciated your comments and suggestions. Accordingly, we have added these contents and quoted the two suggested refs (i.e., ref13 and ref21) in our revised manuscript.

Special thanks to your good comments!
